# Costa Rican Genotype of *Ehrlichia canis*: A Current Concern

**DOI:** 10.3390/vetsci10050316

**Published:** 2023-04-27

**Authors:** Karla Irigaray Nogueira Borges, Nathalia de Assis Pereira, Daniel Moura de Aguiar, Isis Indaiara Gonçalves Granjeiro Taques, Bruna Samara Alves-Ribeiro, Dirceu Guilherme de Souza Ramos, Ísis Assis Braga

**Affiliations:** 1Veterinary Medicine College, Basic Unit of Bioscience, Mineiros University Center, 22nd Street s/n, Mineiros 75833-130, GO, Brazil; karla@unifimes.edu.br; 2Virology and Rickettsioses Laboratory, Department of Clinical Veterinary Medicine, Veterinary Medicine College, Federal University of Mato Grosso, Fernando Correa da Costa Avenue 2367, Cuiabá 78060-900, MT, Brazil; nathaliaassis89@gmail.com (N.d.A.P.); danmoura@ufmt.br (D.M.d.A.); 3EDUVALE College—Campus Jaciara, Caiçara Street, 2.114, Jaciara 78820-000, MT, Brazil; isisgtaques@gmail.com; 4Laboratory of Veterinary Pathology and Parasitology, Academic Unit of Agricultural Sciences, Federal University of Jataí, Jataí 75801-615, GO, Brazil; brunasamara@discente.ufj.edu.br (B.S.A.-R.); dguilherme@ufj.edu.br (D.G.d.S.R.)

**Keywords:** *Ehrlichia canis*, BrTRP36, CRTRP36, USTRP36, immune adaptation, zoonotic potential

## Abstract

**Simple Summary:**

Clinical and hematological characteristics of the blood of 125 dogs with different *Ehrlichia canis* genotypes were analyzed. We observed that the Brazilian genotype is more prevalent and adapted in dogs in the central-western region of Brazil, whereas the American genotype appears to be more pathogenic, causing inflammatory signs. However, we highlight the high frequency of dogs reactive to the Costa Rican genotype, which is suggested to be less adapted to the immune response of the animals. In addition, this genotype has an imminent risk because of its zoonotic potential.

**Abstract:**

Canine monocytic ehrlichiosis (CME) is endemic to Brazil, and studies have verified that dogs have been exposed to different genotypes of *Ehrlichia canis*. This genetic divergence can influence the clinical response of the animals. We aimed to describe clinical and hematological changes in 125 dogs that reacted to BrTRP36, USTRP36, and CRTRP36 genotypes through enzyme immunoassays and to highlight the current concern regarding infection by the Costa Rican genotype. The results showed that 52.0% reacted to the Brazilian genotype, 22.4% reacted to the Costa Rican genotype, and 16.0% reacted to the American genotype, and some co-reactions were observed. Dogs reactive to BrTRP36 were 1.24% more likely to present with medullary regeneration in cases of anemia and 3% less likely to manifest hyperproteinemia, while dogs reactive to CRTRP36 were 0.7% less likely to present with medullary regeneration. Febrile illness and neurological alterations were also statistically associated, with an 85.7% and 231.2% increased likelihood, respectively, to occur in dogs that reacted to USTRP36. The dogs with the American genotype developed clinical manifestations related to systemic inflammation, while those with the Brazilian genotype of *E. canis* were more dispersed in the region studied, showing greater adaptation to the hosts. We highlight the significant serocurrence of the Costa Rican genotype, which has already been described to have zoonotic potential and which showed less adaptation.

## 1. Introduction

Canine monocytic ehrlichiosis (CME) is an endemic disease in Brazil, transmitted by *Rhipicephalus sanguineus* sensu lato ticks [1] associated with a complex symptomatology that varies according to the intensity and clinical stage of the disease [2]. New research has verified that infection by distinct *Ehrlichia canis* genotypes may lead to variations in the animal’s clinical response [3].

Currently, four genotypes of *E. canis* have been identified based on the divergence of the tandem repeat protein (TRP) sequences: USTRP36, BrTRP36, CRTRP36, and more recently, CUBTRP36—the American, Brazilian, Costa Rican [4,5,6] and Cuban genotypes, respectively [7]. TRPs are known to play important roles in the entry of *E. canis* into the host cell, modulation of replication mechanisms, and facilitation of the exit of the bacterium from the infected cell. Thus, they are essential for host adaptation and immune response evasion [8].

In Brazil, the Brazilian and American genotypes are widely distributed, whereas the Costa Rican genotype was observed in 0.6% of the exposed dogs [9]. In a previous analysis, Taques et al. [3] found some correlations between animals exposed to USTRP36 and BrTRP36 genotypes with common changes in CME; however, gaps remain. Thus, further evaluation of the clinical behavior of dogs with different genotypes is necessary, specially due to the lack of data about Costa Rican genotype. We aimed to described clinical and hematological changes in different genotypes of *E. canis* compared to CRTRP36 and to highlight the current concern regarding infection by the Costa Rican genotype.

## 2. Materials and Methods

From March 2021 to June 2022, dogs at veterinary clinics in the city of Mineiros, in the midwestern region of Brazil, were selected for inclusion in the study based on the following criteria: a presence of hematological changes associated with CME (anemia and thrombocytopenia), presence of *Ehrlichia* spp. morulae in blood smears and/or reactivity to SNAP4Dx^®^ Plus Test (IDEXX Laboratories ©, Fremont, CA, USA). Thus, 125 dogs were selected with owner authorizing the use of clinical and hematological data of animals in the research by filling out the informed consent form (ICF). The study was approved by the Ethics Committee on Animal Research (CEUA) of the University Center of Mineiros (protocol number 0088/2020) for biological sample manipulation.

Clinical data, such as age, sex, breed, presence of ticks, body temperature, mucosal evaluation, palpation of lymph nodes and spleen, and data on parameters such as the presence of gastroenteric, hemodynamic, neurological, and ophthalmic alterations, were collected. Additionally, hematological data, such as hematometry, leukometry, and platelet count, were verified, considering the reference values of Meyer and Harvey [10] and Jain [11]. To evaluate medullary regeneration, the presence and intensity of anisocytosis (RDW), polychromasia, and metarubrites was considered.

Subsequently, the canine serum samples were subjected to the enzyme immunoassay technique described by Aguiar et al. [12] using synthetic peptides (Bio-Synthesis Inc^®^, Lewisville, TX, USA) of *E. canis*, with their respective diagnostic targets—TRP19 (protein): HFTGPTSFEVNLSEEEKMELQEVS (synthetic peptide; *E. canis* (genotype) [13]; TRP36US: TEDSVSAPATEDSVSAPA, the American *E. canis* [4]; TRP36Br: ASVVPEAEASVVPEAEASVVPEAE, the Brazilian *E. canis* [12]; and TRP36CR: EASVVPAAEAPQPAQQTEDEFFSDGIEA, the Costa Rican *E. canis* [14]. The optical density cut-off point was 0.250 [12].

Clinical and hematological data were classified as qualitative variables, compared to the presence or absence of each genotype and were subjected to the chi-square test of association using a 95% confidence interval (CI). Statistical significance was set at *p* < 0.05. Statistical analyses were performed using the Epi info vs. 7.2.5.0 software. This analysis was conducted to demonstrate the main alterations in *E. canis* genotypes.

## 3. Results

Of the 125 dogs, 77.6% reacted to the TRP19 genotype, a highly conserved protein in the *E. canis* species, 52.0% reacted to the Brazilian genotype, 22.4% reacted to the Costa Rican genotype, and 16.0% reacted to the American genotype. Co-reactions were also observed (Table 1). The optical densities ranged from 0.258 to 1.768 (mean, 0.834) for the TRP19 peptide, 0.251 to 1.302 (mean, 0.691) for TRP36Br, 0.273 to 1.762 (mean, 1.061) for TRP36CR, and 0.254 to 1.193 (mean, 0.691) for TRP36US.

The racial characteristics, sex, and age range of the seroreactive dogs are shown in Table 2.

The hematological evaluations of the dogs are shown in Table 3. Clinical data are presented in Table 4.

## 4. Discussion

Corroborating what was described by Taques [9], it was found that the genotypes of *E. canis* are widely distributed in the population of dogs in Brazil; however, we highlight the relative frequency of exposure of dogs to the Costa Rican genotype in the studied area, as previously, this genotype had been reported only in the southeast and northwest regions of the country at a low frequency.

It is believed that the bacterium is undergoing a process of genetic recombination through these proteins, which have already been characterized as targets for host-induced immune pressures [5,15,16], seeking means of evasion of the immune response, and even reaching new hosts. Navarrete [7] reported that the genotypic diversity of *E. canis* was associated with the emergence of strains with greater virulence. In this study, co-reaction with *E. canis* genotypes was observed in 84% of the samples evaluated (Table 1), which is suggestive of co-infection by the different genotypes. This mechanism is commonly present in microorganisms and is considered one of the main drivers of genetic diversity in obligate intracellular microorganisms [17].

Bouza-Moura et al. [6] found that the TRP of the Costa Rican genotype is very similar to that of the Brazilian genotype, with the addition of some amino acids, suggesting a possible mutation. Another noteworthy factor is the association of the CRTPR36 genotype of *E. canis* with infections in humans, which has been considered to have zoonotic potential [6]. It was observed that 28.0% of the dogs reacted only to TRP19, suggesting that unidentified genotypes of *E. canis* may be present in the region, such as the most recently described Cuban genotype (CUBTRP36) or even other unreported recombinations.

When analyzing the characteristics of the animals and their respective clinical conditions, it was observed that adult dogs were predominantly associated with TRP19, indicating that this age group was 310.5% more likely to be exposed to *E. canis* than puppies were, especially of the Brazilian genotype (Table 2). Although Harrus and Waner [18] stated that there is no age predisposition to CME, we know that adult animals have longer exposure to factors associated with *E. canis* transmission and can often act as carriers of the bacterium. As for the racial characterization, it was verified that dogs of defined breeds have a 95.5% higher chance of being exposed to *E. canis*, which is in agreement with the literature [19,20], but there were no associations with the other genotypes, and TRP19 was not correlated with sex.

Regarding the clinical and laboratory manifestations of the different genotypes, it was possible to infer that the Brazilian genotype is more adapted to the animals and to the vector *R. sanguineus* s.l. (currently known as *Rhipicephalus linnaei*) [21], as dogs exposed to the Brazilian genotype presented 50.37% fewer chances of being parasitized by ticks at the time of consultation, and these dogs were parasitized throughout their lives by infected ticks. Furthermore, determining the vector competence of species and strains of *R. saguineus* s.l. for each *E. canis* genotype requires further studies.

Moreover, dogs reactive to BrTRP36 were 1.24% more likely to present with medullary regeneration in cases of anemia, suggesting a better response to infection, and 3% less likely to manifest hyperproteinemia, which demonstrates less antigenic stimulation caused by the presence of the bacteria and consequently a lower production of gamma globulins [18]. Taques et al. [3] described a lower correlation between BrTRP36 and hyperproteinemia in another location in central-western Brazil.

Dogs exposed to the American genotype presented an 85.7% greater likelihood of developing febrile illness, on top of neurological alterations such as convulsions and motor incoordination, which were also statistically associated, with a 231.2% greater likelihood of occurrence in dogs that reacted to USTRP36, demonstrating that it is a more pathogenic and immunogenic genotype than the Brazilian genotype, which supports the findings of Taques et al. [9].

For the first time, an analysis of dogs reacting to the Costa Rican genotype, which was proven to be less adapted to the animal, could be performed, during which it was observed that dogs presented a 0.7% decreased likelihood of medullary regeneration and that the average optical density was higher than that of the other genotypes studied.

## 5. Conclusions

The results of this study allowed us to conclude that there are some phenotypic differences associated with the different *E. canis* genotypes. Medullary regeneration was more closely related to the Brazilian genotype and less closely related to the Costa Rican genotype. The TRP19 genotype was associated with hyperproteinemia. Moreover, it was observed that dogs exposed to the American genotype developed clinical manifestations related to systemic inflammation, such as fever and neurological alterations, which in cases of canine monocytic ehrlichiosis occur in an immune-mediated manner. The Brazilian genotype of *E. canis* was more dispersed in the region studied, showing greater adaptation to the hosts. Finally, we highlight the significant serocurrence of the Costa Rican genotype, which showed less adaptation to dogs, and highlight concerns about the possible zoonotic potential attributed to this genotype.

## Figures and Tables

**Table 1 vetsci-10-00316-t001:** Number of dogs that reacted concomitantly to TRP19, BrTRP36, CRTRP36, and USTRP36 peptides from *Ehrlichia canis* from Mineiros city, midwestern region of Brazil.

Peptides	Animals (*n* = 125)	
	Quantity	%
TRP19, TRP36Br, TRP36 CR e TRP36US	1	0.80
TRP19 e TRP36Br	49	39.20
TRP19 e TRP36CR	18	14.40
TRP19 e TRP36US	13	10.40
TRP36Br e TRP36CR	16	12.80
TRP36Br e TRP36US	6	4.80
TRP36US e TRP36CR	2	1.60
Total	105	84

**Table 2 vetsci-10-00316-t002:** Profile of dogs reacting to TRP19, BrTRP36, CRTRP36, and USTRP36 peptides of *Ehrlichia canis* from Mineiros city, midwestern region of Brazil.

Profile	Animals (*n* = 125)					
	Positive	%	TRP19	TRP36 BR	TRP36 CR	TRP36 US
Female	62	49.6	49	32	12	8
Male	63	50.4	48	33	16	12
Adults	98	78.4	78 *	55 ^†^	23	12
Puppy	27	21.6	19	10	5	8
Defined Race	79	63.2	66 ^+^	40	15	16
SRD	46	36.8	31	25	13	4

* *p* = 0.010 (*odds ratio* = 4.105). ^†^
*p* = 0.006 (*odds ratio* = 5.5). ^+^
*p* = 0.024 (*odds ratio* = 1.955).

**Table 3 vetsci-10-00316-t003:** Hematological analyses of the blood of dogs reacting to the peptides TRP19, BrTRP36, CRTRP36, and USTRP36 of *Ehrlichia canis*, from Mineiros city, midwestern region of Brazil.

Alteration	Animals (*n* = 125)					
	Positive	%	TRP19	TRP36 BR	TRP36 CR	TRP36 US
Thrombocytopenia	114	91.2	89	61	27	19
Anemia	96	76.8	74	54	20	14
Medullar regeneration	73	58.4	57	44 *	10 ^†^	11
Hyperproteinaemia	59	47.2	49 ^+^	32 °	15	7
Activated monocytes	38	30.4	30	20	7	6
Leukopenia	32	25.6	25	17	8	5
Leucocytosis	22	17.6	14	13	2	3
Presence of morulae on smear	11	8.8	7	5	2	1

* *p* = 0.028 (*odds ratio* = 2.239). ^†^
*p* = 0.005 (*odds ratio* = 0.299). ^+^
*p* = 0.00003 (*odds ratio* = 1.021). ° *p* = 0.024 (*odds ratio* = 0.97).

**Table 4 vetsci-10-00316-t004:** Clinical manifestation of dogs reacting to the peptides TRP19, BrTRP36, CRTRP36, and USTRP36 of *Ehrlichia canis* from Mineiros city, midwestern region of Brazil.

Clinical Sign	Animals (*n* = 125)					
	Positive	%	TRP19	TRP36 BR	TRP36 CR	TRP36 US
Anorexia	112	89.6	86	60	26	16
Apathy	73	58.4	54	45	17	11
Fever	55	44	39	27	13	13 *
Dehydration	55	44	45	28	8	9
Mucosal pallor	54	43.2	41	32	6	11
Presence of ticks	37	29.6	28	16 ^†^	6	8
Vomit	15	12	14	9	5	0
Diarrhea	15	12	11	8	2	3
Lymphadenomegalia	11	8.8	9	5	4	1
Epistaxis	10	8	10	11	3	3
Neurological changes	9	7.2	9	5	3	4 ^+^
Ophthalmic changes	8	6.4	8	3	2	1

* *p* = 0.026 (*odds ratio* = 1.857). ^†^
*p* = 0.038 (*odds ratio* = 0.496). ^+^
*p* = 0.047 (*odds ratio* = 3.3125).

## Data Availability

Data sharing not applicable.

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
