# Peer review of "Costa Rican Genotype of Ehrlichia canis: A Current Concern"

_vetsci, 2023, doi:10.3390/vetsci10050316_

Round 1
Reviewer 1 Report
This communication reports on epidemiology and clinical signs of for 4 genotypes of Ehrlichia canis in dogs examined in veterinary clinics from Mineiros, Brazil. The study is of interest from a local point of view as new epidemiological and clinical data are provided; whether these data are of enough quality to be published in Vet Sci is more questionable. In any event, there are several issue that require clarification:
1. The goal(s) of the study is (are) not clearly stated. The title emphasizes the finding of the Costa Rican genotype, whereas the last paragraph of the Introduction, states that: “In a previous analysis, Taques et al. [3] found some correlations between animals exposed to USTRP36 and 53 BrTRP36 genotypes with common changes in CME; however, gaps remain. Thus, further evaluation of the clinical behavior of dogs with different genotypes is necessary”. It is unclear what gaps are the authors referring to, why a further evaluation of clinical signs are required, and what the relationship is of these issues with the importance of the Costa Rican genotype.
2. 3. Biases related to the sampling scheme should be carefully considered, and discussed. The epidemiological data refer to dogs brought to vet clinics: are they a representative sample of the dogs inhabiting the locality? In addition, the dogs were brought to several vet clinics and, therefore, ‘clinic’ works as a random factor that cannot be ignored (e.g., it could be a ‘clinic’ effect if there are differences in the efficacy in detecting and describing symptoms, and in doing so in a consistent manner).
2. It is unclear to me what is compared through statistical analysis. There is mention to Chi-square tests, but no the classes to be compared and why. The Tables do not clarify the issue further: are you comparing the frequency of a genotype, clinical sign, etc., between sexes? Between age classes? Why is this important? We need more background to understand the criteria used to select the comparisons; otherwise, you could run the risk of embarking in a “fishing” practice (i.e., comparing everything to uncritically spot significant differences). In any event, you are dealing with multiple, non-independent comparisons. For instance, if you compare the frequency of each genotype between age classes, you are dealing with it for comparisons, and it is clear that some genotypes co-occur in the same dogs, i.e., observations are not independent in the overall test (although the test is chucked in four pieces, one per genotype). At least, a correction of type I error via, e.g., sequential Bonferroni, is peremptory, although it would be better, by far, to deal with all categories with multivariate analyses (see the Paragraph 4).
3. Given that dogs can be positive for several genotypes, it is not wise to ignore this circumstance in the descriptive part, the analysis and interpretation. First, all the tables should consider groups with single, double, triple,… genotypes, so that each group contains independent dogs. This should allow the reader to know how many dogs are in which category. Moreover, it is not correct to compare groups based on E. canis genotypes if some dogs has been infected with more than one of them. For instance, one can wonder whether some clinical effects has to do with the interaction of genotypes; i.e., effects of each genotype are not necessarily additive, and this should be discussed.
4. A more correct way to compare hematological parameters, and clinical symptoms, between genotypes, sexes, ages, or breeds, is with through the use of groups defined by single genotypes, or combinations of them. Then you should compare frequencies of all categories between sexes, ages or breeds. With regard to hematological parameters, and clinical symptoms, each individual dog should be labelled according to its groups (i.e., Costa Rican only, or Costa Rican + American, or American only, etc.), as noted before, as well as other categories (age class, sex, breed). Then, two presence / absence matrix, one containing hematological parameters, other with clinical symptoms, should be built, and subject to Permutation Analysis of Variance (PERMANOVA) to look for different profiles between genotype groups, ages, sexes... If significant differences were found, a subsequent SIMPER analysis could identify which hematological parameters, and clinical signs, contribute more to differences between groups.
You may want to have a look to the package PERMANOVA for Primer to perform these analyses. The myriad pairwise comparisons based on on-independent samples, as shown in the present version, are simply wrong.
5. Please clarify the goals, and analyse the data accordingly, to get a more focused discussion. Once these issues are improved, I will be happy to have a second look at the Ms.
Author Response
This communication reports on epidemiology and clinical signs of for 4 genotypes of Ehrlichia canis in dogs examined in veterinary clinics from Mineiros, Brazil. The study is of interest from a local point of view as new epidemiological and clinical data are provided; whether these data are of enough quality to be published in Vet Sci is more questionable. In any event, there are several issue that require clarification:
1. The goal(s) of the study is (are) not clearly stated. The title emphasizes the finding of the Costa Rican genotype, whereas the last paragraph of the Introduction, states that: “In a previous analysis, Taques et al. [3] found some correlations between animals exposed to USTRP36 and 53 BrTRP36 genotypes with common changes in CME; however, gaps remain. Thus, further evaluation of the clinical behavior of dogs with different genotypes is necessary”. It is unclear what gaps are the authors referring to, why a further evaluation of clinical signs are required, and what the relationship is of these issues with the importance of the Costa Rican genotype.
R: In the end of introduction we added “Thus, further evaluation of the clinical behavior of dogs with different genotypes is necessary, specially the lack of data about Costa Rican genotype. We aimed to de-scribed clinical and hematological changes in different genotypes of E. canis comparing to CRTRP36.”.
The same was added in abstract “We aimed to described clinical and hematological changes in 125 dogs that reacted to BrTRP36, USTRP36, and CRTRP36 genotypes through enzyme immunoassays.”
2.3. Biases related to the sampling scheme should be carefully considered, and discussed. The epidemiological data refer to dogs brought to vet clinics: are they a representative sample of the dogs inhabiting the locality? In addition, the dogs were brought to several vet clinics and, therefore, ‘clinic’ works as a random factor that cannot be ignored (e.g., it could be a ‘clinic’ effect if there are differences in the efficacy in detecting and describing symptoms, and in doing so in a consistent manner).
R: Primarily, there was a failure to describe the objective of the study. However, this has been corrected. We do not aim to analyze prevalence for population inferences. We aimed to evaluate animals naturally infected by E. canis. Therefore, we do not consider clinical animals as a bias, but as a target group of this study.
2. It is unclear to me what is compared through statistical analysis. There is mention to Chi-square tests, but no the classes to be compared and why. The Tables do not clarify the issue further: are you comparing the frequency of a genotype, clinical sign, etc., between sexes? Between age classes? Why is this important? We need more background to understand the criteria used to select the comparisons; otherwise, you could run the risk of embarking in a “fishing” practice (i.e., comparing everything to uncritically spot significant differences). In any event, you are dealing with multiple, non-independent comparisons. For instance, if you compare the frequency of each genotype between age classes, you are dealing with it for comparisons, and it is clear that some genotypes co-occur in the same dogs, i.e., observations are not independent in the overall test (although the test is chucked in four pieces, one per genotype). At least, a correction of type I error via, e.g., sequential Bonferroni, is peremptory, although it would be better, by far, to deal with all categories with multivariate analyses (see the Paragraph 4).
4. A more correct way to compare hematological parameters, and clinical symptoms, between genotypes, sexes, ages, or breeds, is with through the use of groups defined by single genotypes, or combinations of them. Then you should compare frequencies of all categories between sexes, ages or breeds. With regard to hematological parameters, and clinical symptoms, each individual dog should be labelled according to its groups (i.e., Costa Rican only, or Costa Rican + American, or American only, etc.), as noted before, as well as other categories (age class, sex, breed). Then, two presence / absence matrix, one containing hematological parameters, other with clinical symptoms, should be built, and subject to Permutation Analysis of Variance (PERMANOVA) to look for different profiles between genotype groups, ages, sexes... If significant differences were found, a subsequent SIMPER analysis could identify which hematological parameters, and clinical signs, contribute more to differences between groups.
R: Considering this suggestion we changed the statistical paragraph to “Clinical and hematological data were classified as qualitative variables and compared to presence or absence of each genotype and were subjected to the chi-square test of association using a 95% confidence interval (CI). Statistical significance was set at P < 0.05. Statistical analyses were performed using the Epi info vs 7.2.5.0 software. This analysis was conducted to demonstrate the main alterations in E. canis genotypes.”.
Our objective with this analysis was not to compare groups of animals, but what are the main risk factors and/or alterations found in animals with each genotype. Just this. For multiple analyzes or even group analyses, we would need non-reactive (negative) animals, which is not part of the design or purpose of the study.
3. Given that dogs can be positive for several genotypes, it is not wise to ignore this circumstance in the descriptive part, the analysis and interpretation. First, all the tables should consider groups with single, double, triple,… genotypes, so that each group contains independent dogs. This should allow the reader to know how many dogs are in which category. Moreover, it is not correct to compare groups based on canis genotypes if some dogs has been infected with more than one of them. For instance, one can wonder whether some clinical effects has to do with the interaction of genotypes; i.e., effects of each genotype are not necessarily additive, and this should be discussed.
R: As previously mentioned, the design does not favor analysis by groups. For this analysis we would need homogeneous groups (same number of animals, same physical characteristics such as sex and age). This would only be possible with experimental infections, not considering natural infections.
Example: How to compare the multiple infection group (n=1) with the BRTRP39 group (n=49)? This is not possible.
5. Please clarify the goals, and analyse the data accordingly, to get a more focused discussion. Once these issues are improved, I will be happy to have a second look at the Ms.
R: We appreciate every suggestion and hope to have answered or explained the misunderstood parts.
Reviewer 2 Report
The work “Costa Rican genotype of Ehrlichia canis: a current concern” contributes to the understanding of the agent's genotypes and their relationship with clinical conditions in the city under study. It is well written and needs a little organization in relation to the journal's norms.
In the introduction, I think it is important to describe the transmission of the agent by its vector/the tick. In addition, the discussion about the group of the same species of ticks and their potential capacities for transmission of genotypes and region of activity should still be the object of other works.
In line 145 “it was possible to infer that the Brazilian genotype is more adapted to the animals and to the vector Riphicephalus sanguineus sensu lato”. Here, I think that information about ticks was not studied in this work and, therefore, it cannot be affirmed. Here it would be interesting to know about the anamnesis if there are imported animals or people related who have international transit.
Author Response
The work “Costa Rican genotype of Ehrlichia canis: a current concern” contributes to the understanding of the agent's genotypes and their relationship with clinical conditions in the city under study. It is well written and needs a little organization in relation to the journal's norms.
In the introduction, I think it is important to describe the transmission of the agent by its vector/the tick.
R: We are very grateful each suggestion. We changed the first line of introduction to “Canine monocytic ehrlichiosis (CME) is an endemic disease in Brazil, transmitted by ticks Rhipicephalus sanguineus sensu lato [1]”.
In addition, the discussion about the group of the same species of ticks and their potential capacities for transmission of genotypes and region of activity should still be the object of other works. In line 145 “it was possible to infer that the Brazilian genotype is more adapted to the animals and to the vector Rhipicephalus sanguineus sensu lato”. Here, I think that information about ticks was not studied in this work and, therefore, it cannot be affirmed. Here it would be interesting to know about the anamnesis if there are imported animals or people related who have international transit.
R: We added “Furthermore, the vector competence of species and strains of R. saguineus s.l. for each E. canis genotype, requiring further studies.”.